# Multidomain Simulation Model for Analysis of Geometric Variation and Productivity in Multi-Stage Assembly Systems

**Sergio Benavent Nácher \*** , **Pedro Rosado Castellano, Fernando Romero Subirón and José V. Abellán-Nebot**

Department of Industrial Systems Engineering and Design, Universitat Jaume I. Av. Vicent Sos Baynat, s/n 12071 Castellón de la Plana, Spain; rosado@uji.es (P.R.C.); fromero@uji.es (F.R.S.); abellan@uji.es (J.V.A.-N.)
\* Correspondence: benavens@uji.es

**Abstract:** Nowadays, the new era of industry 4.0 is forcing manufacturers to develop models and methods for managing the geometric variation of a final product in complex manufacturing environments, such as multistage manufacturing systems. The stream of variation model has been successfully applied to manage product geometric variation in these systems, but there is a lack of research studying its application together with the material and order flow in the system. In this work, which is focused on the production quality paradigm in a model-based system engineering context, a digital prototype is proposed to integrate productivity and part quality based on the stream of variation analysis in multistage assembly systems. The prototype was modelled and simulated with OpenModelica tool exploiting the Modelica language capabilities for multidomain simulations and its synergy with SysML. A case study is presented to validate the potential applicability of the approach. The proposed model and the results show a promising potential for future developments aligned with the production quality paradigm.

**Keywords:** production quality paradigm; stream of variation; multi-domain modelling and simulation; SysML; Modelica; multistage assembly system

## 1. Introduction

The industries, as they advance in the postulates advocated in different research and development roadmaps, e.g., industry 4.0, are transforming their factories and their management and control strategies following some of the emerging manufacturing paradigms such as smart manufacturing or production quality.

On the one hand, production quality was formulated as a paradigm that combines quality, production logistics, and maintenance methods and tools to maintain the throughput and the service level of conforming parts under control and to improve them over time, with minimal waste of resources and materials [1]. One of the main issues in production quality is the variation propagation modelling for quality control, process monitoring, and root cause identification for multistage manufacturing systems (MMS). These manufacturing systems are highly complex, and the development of an effective and cost-efficient quality control strategy that intermeshes and links closed-loop quality control systems at various levels of the company is crucial [1]. A clear example can be found in [2], where different control strategies are proposed using newly available sensor data from shopfloors to prevent the generation and propagation of defects throughout the process.

In the literature, different models for variation propagation analysis have been studied in MMS. One of these approaches is the so-called stream of variation (SoV) model, proposed in [3] for 2D

multistage assembly processes (sheet metal assembly processes) and later on extended for 3D multistage assembly and machining processes [4,5]. The SoV model has been successfully applied in a large number of applications in MMS, especially for product quality assurance [6], process planning [7], and fault diagnosis [8], as well as applications for decision-making and optimization for inspection planning [8–10] and quality control and error compensation with the use on shop-floor data in real time [11,12].

On the other hand, many research works that have been focused on the smart manufacturing paradigm [13,14] show that these new control, diagnosis, and improvement strategies in MMS should exploit the wide-ranging potentials of cyber-physical systems (CPS) and the digital twin approach in order to support dynamic coordination among the system, the product, and the process throughout their life cycles. For this purpose, it is required to use technologies such as in-line data gathering solutions, interoperability standards, modelling and simulation tools, data analytics, and digital manufacturing technologies. From these technologies, modelling and simulation tools are acquiring great relevance in the smart manufacturing paradigm with investigations focused on concepts such as digital twins, digital shadows, and digital models [13]. From a simulation point of view, the digital twin approach can be seen as the next disruptive technology in modelling, simulation, and optimization technology enabled by the higher capabilities of computer equipment over the years and the adoption of model based system engineering (MBSE) principles by the industry [15].

Furthermore, as the introduction of the principles and practices of the two aforementioned paradigms progresses, it is increasingly important to analyze the dynamic behavior of the overall system under various realistic circumstances. For this reason, the use of simulation in a domain-specific context with domain-specific tools in the engineering of complex systems is no longer a valid solution, moving to other tools with capability of integrating different domains. In this direction, some initiatives promoted by the Modelica Association can be found, which were initially focused on the development of the Modelica language and Modelica libraries [16] to facilitate modelling and simulation of complex systems and, more recently, on the design of CPS. However, despite numerous efforts made in the field of modelling and simulation under these paradigms, most of them mainly focused on production and maintenance analysis with discrete event simulations; there is a lack of research oriented towards the development of frameworks for the management of geometric variations, especially in modelling and simulating the effects of geometric variations for quality assurance, productivity, and their interrelations.

To overcome this limitation, this work proposes the development of a multi-domain and multi-scale modelling and simulation prototype that allows visualizing and evaluating the functionalities and operation modes of multistage assembly systems (MAS), both in terms of geometric quality and productivity. The demonstrator is implemented using Modelica language, an object-oriented language for modelling and simulation of complex systems. The analysis of the logistic flow is based on Discrete Event System Specification (DEVS) formalism, while the geometric quality analysis is conducted under the engineering-driven method defined by the SoV model.

The paper is organized as follows: Section 2 presents the background of the study with the description of the SoV modelling approach for geometric variation propagation and the methodology followed to develop and validate the proposed meta-model of simulation for MAS. Next, Section 3 describes the adopted MAS type for this work, a brief SysML description of the proposed meta-model for the MAS simulation system, and its subsequent implementation in OpenModelica. In order to validate the proposed meta-model, Section 4 presents a case study of a MAS composed of four stages, and different scenarios are analyzed in order to compare potential control logics to improve productivity and quality. Finally, a discussion about the results is provided in Sections 5 and 6 concludes the paper.

## 2. Methodologies and Techniques of the Research Work

This research work is part of a broader research project that aims to design and implement a multi-scale and multi-resolution modelling-simulation platform to enable assessment and control of geometric quality variability and productivity in MMS. Under this wider research context, this work

is focused on the definition of simulation models for modular and flexible manufacturing systems, paying special attention to the design phase on these systems.

Particularly, this work is focused on the development of a demonstrator for modelling and simulating a MAS while considering performance indicators of quality variability and productivity. This section provides an overview of the methodologies and techniques used in our research work. Firstly, the SoV model is briefly described for MAS and the error propagation framework and the main sources of error in this type of MMS are defined. Secondly, the proposed methodology for the design of the meta-model for MAS simulation is described.

### 2.1. Error Propagation Modelling through SoV Model

The SoV model is able to estimate the error propagation throughout a MMS by modelling a series of kinematic relationships between fixtures, datum surfaces used for locating parts within the fixtures, and the deviation of the manufacturing action itself, such as welding, machining, etc. The mathematical formulation is based on the well-known state space model representation commonly used in the field of control systems [17]. For a MAS, the SoV model can be mathematically expressed as [18]:

$$X_k = A_k \cdot X_{k-1} + B_k \cdot U_k + W_k, \tag{1}$$

where $X_k$ is the vector that represents the geometric quality deviations of each part of the assembly after the manufacturing stage, $k$; $X_{k-1}$ represents the same deviations but after stage k−1, which becomes the input of stage k; $U_k$ represents the deviations of the fixture locators that hold the parts during the manufacturing operation; matrices $A_k$ and $B_k$ are obtained through kinematic analyses and represent the reorientation effect of the assembled parts after stage $k-1$ when placed on stage $k$ and the effects of the deviations of the locators over the resulting quality deviations at stage $k$, respectively; $W_k$ represents the modelling errors since matrices $A_k$ and $B_k$ are obtained through kinematic analyses subjected to a certain linearization degree. The geometric quality deviations of each part represented in $X_k$ is defined by the deviation of the local coordinate system (LCS) of the part, since the parts are assumed to be rigid. Thus, the deviation of the LCS is modelled by a differential motion vector (DMV) in the form $[\mathbf{d}, \theta]^T$, where $\mathbf{d}$ and $\theta$ are the position and orientation deviation of the LCS, respectively.

In this multistage process, an inspection stage may be placed along the process for quality control. Assuming an inspection stage after the assembly stage k, the inspection results are described as:

$$Y_k = C_k \cdot X_k + V_k, \tag{2}$$

where $Y_k$ is the vector that represents the deviation of the features inspected after stage $k$ and, therefore, it is a function of the current deviation of the resulting assembly ($X_k$) through the matrix $C_k$ plus the measurement error represented by the vector $V_k$. Both Equations (1) and (2) define the error propagation in MAS of N stages, where $k = 1, \ldots, N$. An example of MAS and the effect of error propagation is shown in Figure 1, where fixturing is based on 4-way and 2-way locators. As it can be seen, a locator error at stage $k$ produces an error in the resulting subassembly after stage $k$. Thus, this subassembly enters stage $k+1$ and, due to relocation error, and even if no fixture or welding error is added, the resulting subassembly after stage $k+1$ presents a geometric error. Later, at the inspection stage, the assembly is inspected and a quality error is identified. Readers interested in a detailed description of SoV model derivation, please refer to references [3–5,18].

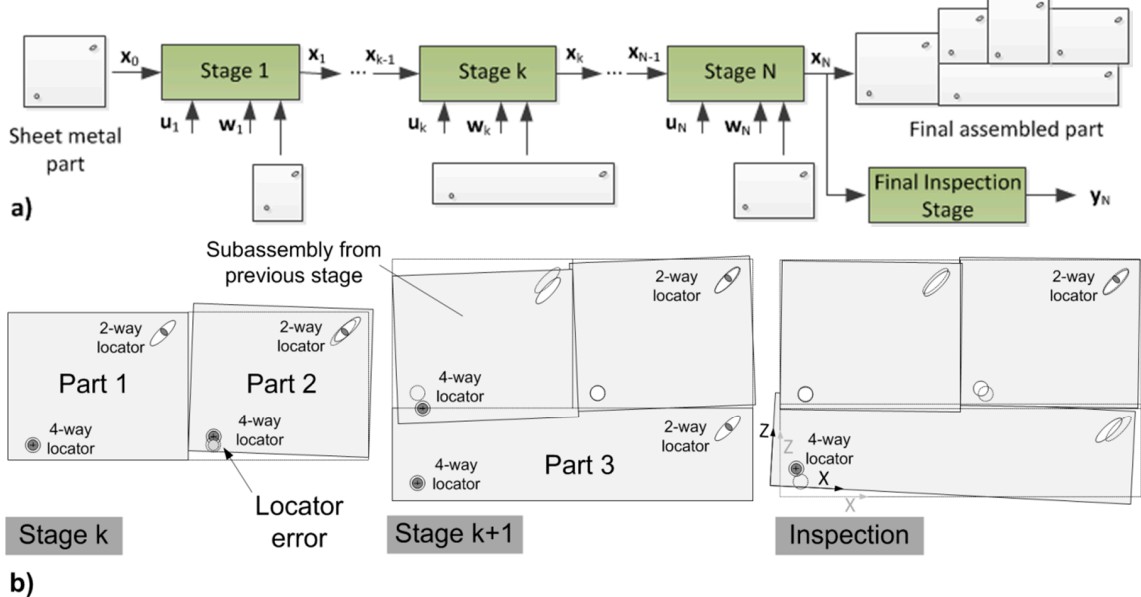

**Figure 1.** (**a**) Multistage assembly system (MAS) with N stations; (**b**) illustration of error propagation in a 2D MAS.

### 2.2. Methodology Adopted to Define the Simulation Model

Since the end of the last century, the methodologies of design and modelling of object-oriented systems have been gaining popularity in the field of systems analysis and design, especially since the development of the unified modelling language (UML) in 1997. Although the initial use of the UML was focused on software systems, it was rapidly extended to other mixed systems (mechatronics, embedded, etc.) and to complex systems (systems of systems), as is the case of the current cyber-physical manufacturing/production systems, by UML profiles as systems modelling language (SysML) [19].

While the use of SysML is crucial for establishing formal and unambiguous communications between system engineers, disciplinary designers, and system analysts, Modelica is an equation-based, object-oriented behavioral simulation language that has been shown to be a suitable language to take into account geometrical and multi-physical constraints at the conceptual design phase under a MBSE approach [20,21]. The synergy created by integrating SysML and Modelica in the design and specification of simulation systems is very significant and the advantages from the joint use of both languages are multiple [22,23]. However, MBSE promotes the construction of executable simulation models (e.g., Modelica models) from descriptive models, such as SysML models. Therefore, there is a need to automate SysML–Modelica transformations and vice versa [24].

The methodology proposed to define the simulation model is based on both SysML and Modelica. Furthermore, the methodology adopts the MBSE approach and it follows the flow of activities for product development proposed in the extended V-model [25], which incorporates an approach to virtual product development. Note that this research work can be placed in the first phases of the V-model related to the process of systems, when the phase for modelling and specification of the system to be analyzed has already been carried out and a design of the concept of the system is given.

The proposed methodology is inspired by the results of some published research [24], mainly in the field of product development, with the aim of systematizing the tasks involved in the design and implementation of simulation systems. The methodology is composed of five steps and it is generally enough to be applied in the definition of any type of simulation model:

1. Analysis of the functional and logical structure of the system to be simulated. The system must be analyzed in order to know the structural elements and their relationships that characterize it. This structure is usually described using a SysML block definition diagram (BDD), where the logical system components are represented by blocks and activity diagrams for process modelling.

2. Definition of concepts and requirements for the simulation system model. For this, various aspects must be considered, such as the type of simulations (e.g., continuous dynamic or discrete events), the goals of the study (e.g., maximum defect index value, minimum throughput value, etc.), and the simulation alternatives for the prototype. For this task, SysML diagrams, such as BDD and requirement diagrams are used.

3. Structural and behavioral modelling of the simulation system (the system of interest). Taking into account the results of the previous step, a viable solution is proposed for the objects structure of the simulation system and their prescribed behavior. Structure and behavior descriptions are decomposed to the level of abstraction required and they are described using SysML diagrams, such as BDDs, internal block diagrams (IBDs), parametric diagrams, activity diagrams, sequence diagrams, etc.

4. Modelling the simulations. In this step, the simulation system is related to other relevant design information through: (a) defining a model context, where the elements of the simulation system are bound to the corresponding elements of the system to be analyzed in order to preserve the consistency among all models; (b) modelling the simulations, by defining the simulation runs to be performed; (c) abstracting the simulation by defining the measure of effectiveness from simulation; (d) using the abstracted simulation in the context of design optimization by relating them to stakeholder requirements and measures of effectiveness pointed out in the model of the system to be analyzed (Step 1).

5. Transformation of SysML models into Modelica and validation of the meta-model through the use of the meta-model on a case study. To obtain an executable simulation model, the SysML description of the system must be implemented using a language with this capability, such as Modelica.

## 3. Modelling and Implementation of the Simulation Prototype

### 3.1. The Assembly System to Be Analysed and the Simulation System Requirements

The first step of the methodology is to define the logical structure of the system to be analyzed. We focused our work on a type of assembly system (AS) that has many of the properties (flexibility, reconfigurability, reactivity, intelligence, etc.) that should be present in advanced manufacturing systems. This type of AS, named the flexible and modular assembly line [26], is characterized by: (a) adopting a mixed production model offering flexible routes of the products throughout the system, (b) using automatic guided vehicles (AGVs) to provide it with the great flexibility required, and (c) adhering to a whole set of additional design principles. Among these principles, the need for flexible integration of quality control loops stands out, with some of them focused on preventing the generation and propagation of geometric defects. The objective of this work is to validate these strategies with the construction of a simple simulation prototype supported by a simulation meta-model that will be described in the following sections.

For the sake of simplicity, in this work, we focus on the simulation of a MAS for the assembly of a single product, with an established multi-station process without considering alternative routes, and subject to a number of restrictions that are described throughout this section. The assembly line consists of a certain number (N) of assembly stations to execute the assembling operations and can include inspection stations for the in-line measurement and verification of the final assemblies in order to establish their acceptance or rejection. Each assembly station includes an assembly machine that adds a new part on the incoming subassembly. To execute this operation, the machine previously fixes the part and the subassembly using a set of locators which can be adjusted manually or automatically. Furthermore, the assembly stations may have capabilities for conducting in-process measurements on the resulting subassembly, although the measurements in this case will present greater uncertainty than those obtained from the inspection stations. Both the measured data from the on-line measurements (inspection stations) and from the in-processing measurements (assembly stations) can be used to compute, according to different optimization algorithms, the correction of the

locators' positions for improving the quality index for the final assemblies. Moreover, each assembly station has two input buffers, one for subassemblies and one for parts, to regulate the flow of materials. The transport of processed assemblies between the stations is carried out by means of a set of AGVs, the number of which is sufficient to avoid any stock rupture in the buffers caused by delays or lack of transport capacity. Therefore, in this first work, the transport processes are not modelled.

From the above description, the work is clearly focused on a generic MAS with a sequential linear flow of materials through a variable number of stations which allow the processing of the assemblies and their measurement. This MAS is also equipped with a control logic both for the logistics (flow of materials and production orders) and for the assurance and improvement of geometric quality. Therefore, it is a generic MAS type, which can be described with the modular and scalable structure of the meta-model shown in the BDD of Figure 2.

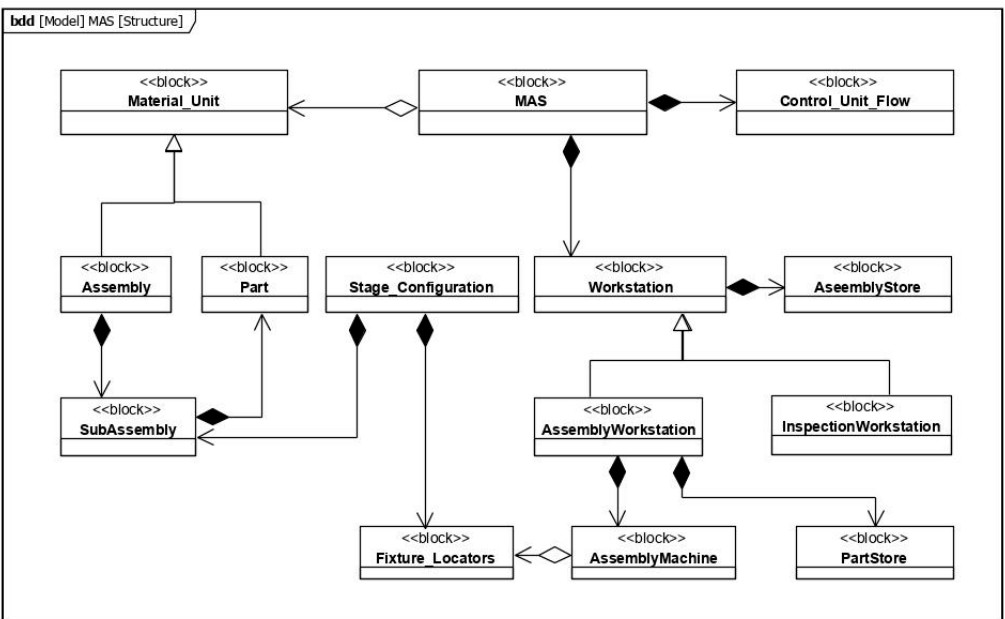

**Figure 2.** Block definition diagram for the generic MAS structure.

Next, and according to the second step of the methodology, the requirements for the system of interest, i.e., the simulation system, are established. Taking into account the previously described characteristics of the system to analyze, the simulation prototype and its meta-model must also be modular and scalable in order to support the creation of specific simulation models for a given MAS. Additionally, the prototype must be designed taking advantage of Modelica's ability to obtain executable models from libraries formed by a set of reusable blocks/classes. These three requirements are fundamental when designing a flexible and modular prototype that, in the future, can be specialized to enrich the simulation by incorporating the behavior models of a smaller scale system, such as multi-axis systems, locator configuration systems with automatic adjustment, etc.

## 3.2. Proposed Simulation Meta-Model

Once the functional and logical structure of MAS and the requirements for the simulation model (Steps 1 and 2) have been defined, the next step is to establish the meta-model for the simulation system of an assembly line. This meta-model must define the static structure of the blocks and the behavior of those blocks, which will serve as constructive elements for the creation of the corresponding executable simulation model for a specific assembly line.

To support the production quality paradigm, the simulation meta-model is structured in two model library packages that include the reusable blocks to compose the simulation models. Specifically, these libraries are: (a) a basic one for the logistics flow simulation, which supports the

data corresponding to order processing, monitoring, control times, etc., and (b) a specialization of the previous one that adds the simulation of the geometric variability propagation according to the SoV technique. This geometric variation flow simulation supports the data corresponding to the differential motion vector of the parts of the assembly and their propagation. The model structuring using these libraries enables the subsequent development of other libraries to simulate the variability propagation according to other techniques, for example, based on data models, or to simulate the system with other planning and production control logics, for example, Kanban. In the rest of the section, in order to not extend too much, only some of the diagrams corresponding to the construction of the second library are presented.

The BDD in Figure 3 shows the static structure of the simulation system, our system of interest. The Simulation_AssemblyProcess block is composed of a number of AssemblyWorkstationProcess blocks, corresponding to the assembly processes executed at the workstations; a number of InspectionWorkstationProcess blocks, corresponding to the inspection processes executed at the inspection stations; an OrderArrivalProcess block, corresponding to the assembly order arrival process; an OrderSinkProcess block, corresponding to the removal of the completed assembly orders. Additionally, the Simulation_AssemblyProcess blockl is composed of a Simulation_Control block that, in addition to monitoring performance measures, such as throughput and defect index, is responsible for reducing the variability of key quality characteristics by using different optimization logic and algorithms to calculate corrections in the position of locators.

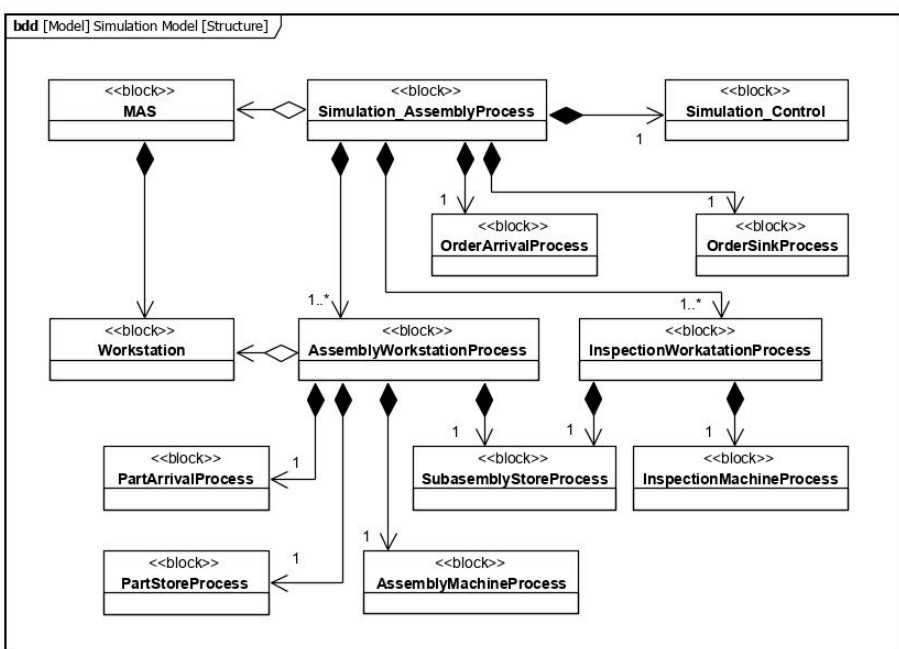

**Figure 3.** Block definition diagram of the simulation system.

Figure 4 shows the internal block diagram of the composed block AssemblyWorktationProcess. As can be seen, each AssemblyWorkstationProcess block consists of two block instances corresponding to the queuing process of the subassemblies (SubassemblyStoreProcess) and the parts (PartStoreProcess); a block instance corresponding to the arrival process of the parts that are incorporated into the processed assembly (PartArrivalProcess); a block instance corresponding to the assembly process (AssemblyMachineProcess). The diagram also shows the three types of connections between the blocks: a) assembly_item, which contains the data corresponding to the assembly flow in the line and its quality characteristics, b) part_item, which contains the data corresponding to the flow of the incoming parts and their quality characteristics, and c) control_data, which contains the data for stations monitoring and for corrections to be applied in the position of locators.

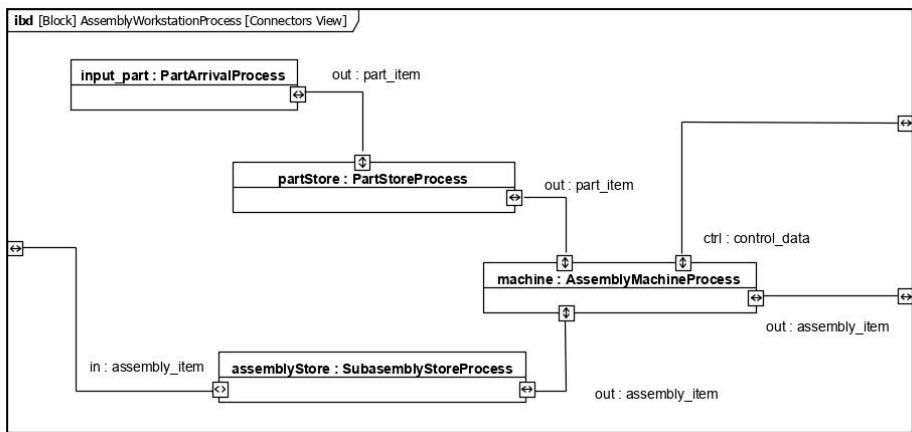

**Figure 4.** Internal block diagram of the WorkstationProcess block.

An important aspect in the definition of the blocks that compose the structure of the simulation system model is their behavior. In this case, the behavior of the system blocks is of a discrete nature and it is modelled using the discrete event systems specification (DEVS) formalism [27]. In DEVS, the behavior of indivisible blocks (e.g., MachineProcess, PartStorageProcess or ControlSimulation) is governed by an atomic DEVS model defined by a set of states, a set of input events, a set of output events, and four functions that define the dynamics: (a) the time advance function, (b) the internal transition function, (c) the external transition function, and (d) the output function. On the other hand, the behavior of compound blocks (e.g., AssemblyWorkstationProcess or the Simulation_AssemblyProcess itself) is governed by a coupled DEVS model, in which output events from one subsystem are input events from another subsystem. The signals corresponding to the input and output events handled in the atomic DEVS model are included in the assembly_item, data_item, and control_data connectors, indicated above.

For the definition of the atomic DEVS models corresponding to the different types of blocks, a categorization structure has been defined. This categorization allows to specialize the blocks from some ancestors (general class), depending on the type of input and/or output events and the type of connectors (ports) used.

As previously indicated, the behavior model of indivisible blocks is governed by an atomic DEVS model. This behavior is represented by a state machine diagram because it begins when the block is instantiated and finishes when the instance is destroyed. This type of behavior can be displayed using a SysML state machine diagram. For example, in Figure 5, the state machine diagram for the Assembly_Process block is shown. As can be seen, the states of the assembly process are: Idle, Setup, Assembling, Repairing, and Blocked. Idle is a composited state of the Empty, Part_Loaded, Assembly_Loaded, and Ready substates. The diagram also shows the transitions between states that are triggered when certain events occur. For example, the transition between Assembling and Repairing is triggered by the failure_event.

Once the simulation system is modelled, and as indicated in Step 4 of the methodology, the modelling of simulations is the next task. In this step, the simulation system model must be related to other relevant design information for defining a simulation. The BDD in Figure 6 defines the model context that shows the relation between the system to be analyzed (MAS) and the simulation system (Simulation_AssemblyProcess). Furthermore, the model context shows that each execution of a simulation (SimulationRun), in addition to using a simulation system (Simulation_AssemblyProcess), contains a set of parameters that define the simulation scenario (Simulation_Scenario). Two types of parameters are distinguished: those that are static in each simulation and only change in the different simulations of a MAS, StaticScenario, and those to which dynamic changes can be assigned during each simulation, DynamicScenario, such as the wear of locators, a change in the failure rates of a machine, etc.

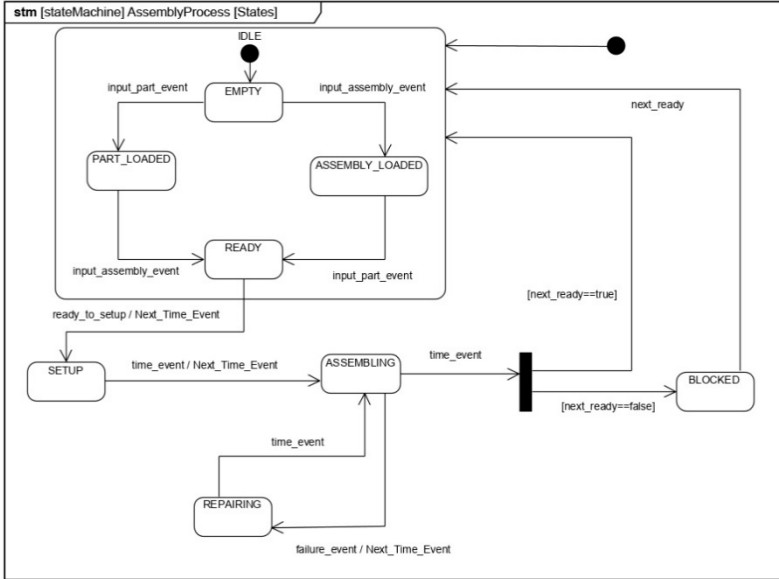

**Figure 5.** State machine diagram for AssemblyProcess block.

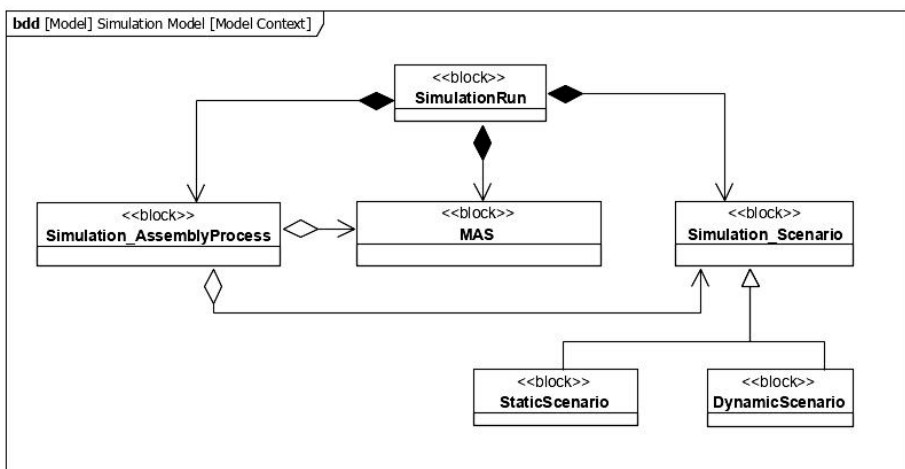

**Figure 6.** Block definition diagram of the analysis context model of the assembly line domain.

The full model specification includes a number of different types of SysML diagrams which, due to space constraints, have not been included in the manuscript.

### 3.3. Model Implementation Using Modelica

This section contains a synthesized description of the implementation of the previously presented SysML meta-model in the Modelica language using mainly textual code to define classes and connectors with sufficient detail for generating executable simulation models. Modelica also supports the graphical appearance definition of components using annotations. This resource provides a powerful mechanism to build complex objects of study, as in the case of the MAS, whose OpenModelica graphical view is shown in Figure 7. OpenModelica software has been used to carry out both the implementation of the reusable libraries expressed with the Modelica language and the construction and execution of the simulation models.

As mentioned before, the meta-model establishes a simulation model of the logistic flow which is later specialized to include the simulation of the quality characteristics flow. In this way, two Modelica packages are modelled: logistic flow simulation (LFS) and variation flow simulation (VFS), developed for the linear MAS described in the previous section. Figure 7 is an OpenModelica screenshot showing the structure of the libraries (packages) and a diagram view of an instantiated model. As can be

appreciated in the figure, different types of Modelica classes have been modelled, mainly connectors, models, and functions. Some of these classes are discussed in greater detail below, but many others cannot be described in detail due to space reasons.

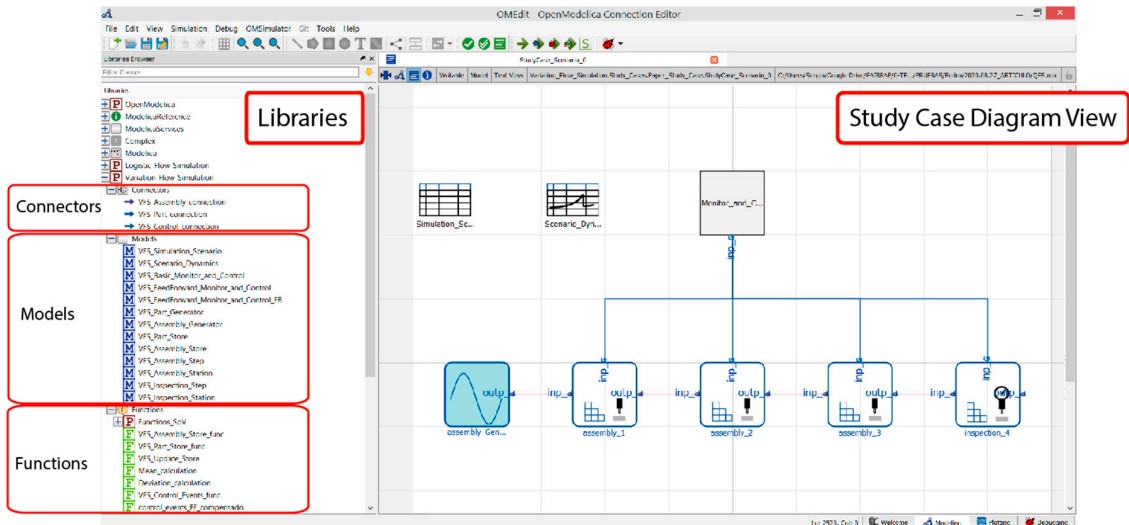

**Figure 7.** OpenModelica screenshot. On the left side: logistic flow simulation (LFS) and variation flow simulation (VFS) libraries, and some VFS internal classes (connectors, models, and functions). On the right side: a diagram view of an instantiated model for the study case.

### 3.3.1. Logistic Flow Simulation

The logistic flow simulation (LFS) package contains the necessary classes to simulate the flow of materials and production orders throughout the stages (assembly and inspection). In a synthesized way, the developed classes for the simulation system have to support different tasks: the initial parameters' establishment (Simulation_Scenario class) and their behavior modelling (Scenario_Dynamics class); the assembly process construction and DEVS formalism through the different stages (Assembly_Station and Inspection_Station classes); data transformation due to the process or control actions (Monitor&Control class), etc. These main classes are described briefly below:

- Simulation_Scenario. This Modelica class contains the initial parameters that define the simulation scenario and corresponds to the Simulation_Scenario block of the SysML model. The main parameters are grouped into the following groups:

  - Assembly_Plan: parameters to collect the sequence of assembly and/or inspection stages. For each stage, the following are indicated: the station in which it is carried out, the subassembly and built-in part, as well as the parameters to model the process time through a normal distribution (mean and a standard deviation).
  - Assembly_Identification: parameters that identify each of the types of subassemblies used in the various stages, indicating the types of part included.
  - Other parameters to model assembly orders' arrival, part arrival, setup time, failure rate, storage capacity, etc.

- Scenario_Dynamics. Some initial parameters defined in Simulation_Scenario can be modelled with continuous functions or discrete changes, introducing a temporal variation. The Scenario_Dynamics class is used to model this dynamic behavior, for example, to consider a gradual increase in the processing time caused by the continuous degradation of the assembly station due to its use, or a punctual (sudden) change in the distribution of the arrival time of the parts due to a change of supplier.

- Assembly_Station and Inspection_Station classes correspond to the WorkstationProcess block from the SysML model and represent an assembly and inspection workstation, respectively.

As described in the SysML model, these complex blocks with a coupled DEVS behavior, in turn, are composed of single blocks which have a discrete behavior definition using an atomic DEVS formalism. All these single blocks use data from Simulation_Scenario and Scenario_Dynamics. In Figure 8, the internal structure of these Modelica classes is shown, mainly composed of the classes described below:

- ○ Part_Generator and Assembly_Generator. These models define the process of generating the arrival of parts and assembly orders, respectively. These Modelica models correspond to PartArrivalProcess and AssemblyArrivalProcess blocks in the SysML model. Part_Generator is represented by a blue box in Figure 8.
- ○ Part_Store and Assembly_Store. These blocks allow for storing input parts or assemblies for a WorkstationProcess. Actually, these stores use a First-In, First-Out strategy but they can be extended to include other delivery strategies. They are represented by red and yellow boxes, respectively, in Figure 8.
- ○ Assembly_Step and Inspection_Step. These blocks correspond to the MachineProcess and InspectionProcess blocks from the SysML model and constitute the process itself characterized by an atomic DEVS formalism. Assembly_Step is represented by a green box in Figure 8.

- Monitor&Control. This Modelica class collects data from each WorkstationProcess, allowing it to execute a statistical analysis to estimate the productivity performance parameters. In addition, this is the basic class that can be specialized to carry out control actions, e.g., a change of the assembly processing order using a dispatching rule.

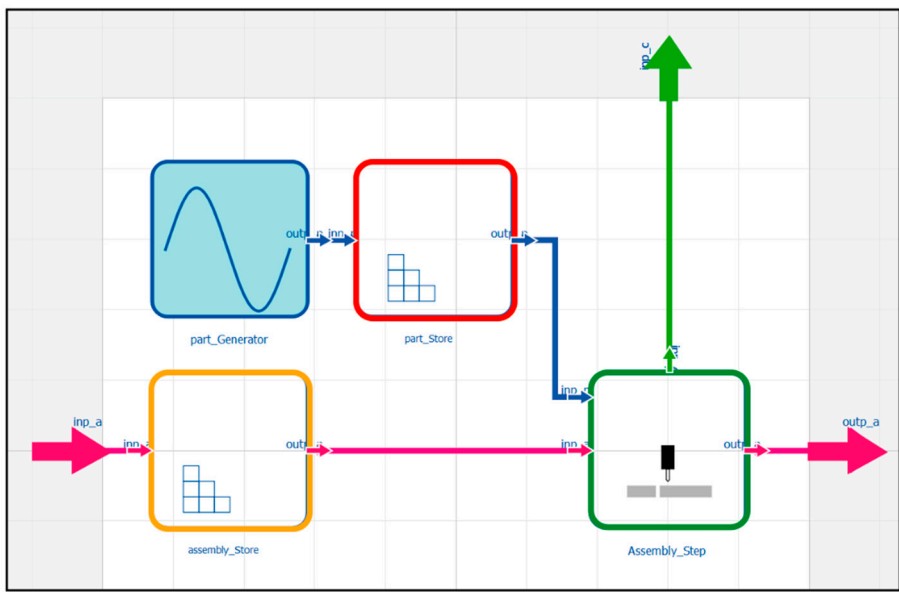

**Figure 8.** Structure of the AssemblyStep (MachineProcess) block in the OpenModelica graphical interface.

Additionally, three types of connectors are defined in Modelica to support different data flows between blocks. These connectors are:

- Assembly_connection. This type of connection contains the data related to the assemblies' flow. These data include the assembly order, the parts contained, and the types of these parts. Additionally, these connectors include Boolean variables to indicate the discrete external events linking the blocks that participate in the assemblies' flow. This connector type is represented by a pink line in Figure 8.

- Part_connection. This type of connection contains the data related to the parts flow that are added in each assembly stage. In this case, these data include the part identification and its type. It also contains the variables to link the blocks that participate in the parts flow. This connector is represented by a blue line in Figure 8.

- Control_connection. This type of connection contains the data used to monitor and control the WorkstationProcess. These connections include data flowing from the WorkstationProcess to Monitor&Control block, such as workstation status, processing time, setup time, and so on. It also contains data that flows from the Monitor&Control block to WorkstationProcess to carry out control actions, such as maintenance stop, setup actions, etc. This connector is represented by a green line in Figure 8.

3.3.2. Variation Flow Simulation Using 2D-SoV

As mentioned above, the variation flow simulation (VFS) library is developed to simulate the quality characteristics flow restricted to 2D geometric characteristics using the SoV technique. For this, LFS classes and connectors previously described are extended to incorporate variables, parameters and functions related to characteristics.

In the VFS library, the LFS connectors are extended with additional data gathering the deviation motion vector (DMV) of the SoV technique that contain the translation and orientation errors and are transmitted together with the rest of the part or assembly data in the connector. Control_connection also includes other variables related to the measured characteristics on each inspection stage and the optimal positions for fixture locators calculated according to different control logics.

LFS classes are also extended as follows:

- VFS_Simulation_Scenario and VFS_Scenario_Dynamics extend the Simulation_Scenario and Scenario_Dynamics classes from the LFS library by incorporating the following parameters:

  ○ Additional process plan data to specify the locating scheme for each assembly or inspection stage. As shown in Figure 1, each part or subassembly is located using a 4-way and a 2-way locator.

  ○ Geometric data that define the nominal parts' geometries and nominal locators' positions and their respective variations modelled by a normal distribution with a mean and a standard deviation. These geometric variations can be modelled as a stable distribution or as a distribution whose parameters change progressively in time. For this, different VFS_Scenario_Dynamics extensions are used.

  ○ Measurement of uncertainties at each stage with a measurement process (assembly stations with in-process measurement capability and inspection stations), modelled by a standard deviation. In-process measurements in assembly stations will present greater uncertainties than those obtained in the inspection stations.

- VFS_Assembly_Station and VFS_Inspection_Station extend the Assembly_Station and Inspection_Station classes from the LFS library to incorporate the necessary properties and functions for computing the DMV transformations in assembly stations and measured quality characteristics in inspection stations and assembly stations with in-process measurement, according to the SoV technique.

Moreover, this library incorporates an additional VFS_Basic_Monitor&Control class which extends the Monitor&Control class from the LFS library and executes the monitoring of the quality characteristic flow. This extension is also specialized to support different quality control loops. Specifically, for this work, the next two VFS_Basic_Monitor&Control specializations have been developed to support alternative control logics:

- VFS_FeedForward_Monitor&Control implements the technique proposed in [10]. This control logic, named feed-forward control, uses measurement data from an early stage to calculate the

optimal position of locators placed at subsequent stages to minimize deviations in the final assembly. It is important to note that this calculation is carried out individually for each assembly order. To apply this logic, it must be also considered that the fixtures of subsequent stages have actuators to adjust the position of each locator. In feed-forward control, from the measurements at stage ($i-1$)

$$y_{i-1} = C_{i-1} \cdot x_{i-1} + V_{i-1}, \tag{3}$$

the goal is to obtain the vector $U_i = \{u_i, u_{i+1}, \dots, u_N\}^T$ of a locator's position that minimizes the expected value of measured deviations at final stage $y_N$. Then, it can be stated as

$$\min_{U_i} E\{(y_N)^T \cdot (y_N)\} = \min_{U_i} E\{(C_N \cdot x_N)^T \cdot (C_N \cdot x_N)\}. \tag{4}$$

using Equations (1) and (2),

$$C_N \cdot x_N = \Gamma_i \cdot x_{i-1} + \Psi_i \cdot U_i + \Omega_i \cdot W_i, \tag{5}$$

where

$$\begin{cases} \Gamma_i = C_N \cdot A_N \cdot \dots \cdot A_i \\ \Psi_i = C_N \cdot [A_N \cdot \dots \cdot A_{i+1} \cdot B_i, A_N \cdot \dots \cdot A_{i+2} \cdot B_{i+1}, \dots, B_N] \\ \Omega_i = C_N \cdot [A_N \cdot \dots \cdot A_{i+1}, A_N \cdot \dots \cdot A_{i+2}, \dots, I] \end{cases}. \tag{6}$$

This minimization problem can be solved to obtain the $U_i$ applying least squares to

$$\Psi_i \cdot U_i = -\Gamma_i \cdot x_{i-1}. \tag{7}$$

Previously, the values of $x_{i-1}$ had to be obtained from the measurements of vector $y_{i-1}$ by applying least squares to Equation (3), considering that the mean value of $V_{i-1}$ is zero.

- VFS_FeedForward&Backward_Monitor&Control extends the previous control class to calculate the optimal position for the locators for all stages using a refined logic. Besides the feed-forward control logic, a backward control logic is added and an additional correction in the current stage is conducted using the deviation mean detected in previous processed assemblies at this stage. In backward control, in addition to the vector $U_{i-1} = \{u_1, u_2, \dots, u_{i-1}\}^T$ is computed to compensate the deviations at stage ($i-1$), taking into consideration the mean values for DMV $\bar{x}_{i-1}$. that are computed from the mean values of last measurements $\bar{y}_{i-1}$, by applying least squares to the expression

$$\bar{y}_{i-1} = C_{i-1} \cdot \bar{x}_{i-1}. \tag{8}$$

This compensation is applied to obtain a final $y_{i-1} = 0$. Therefore,

$$- C_{i-1} \cdot \bar{x}_{i-1} = \Gamma_{i-1} \cdot \bar{x}_o + \Psi_{i-1} \cdot U_{i-1} + \Omega_{i-1} \cdot W_{i-1} \tag{9}$$

where

$$\begin{cases} \Gamma_{i-1} = C_{i-1} \cdot A_{i-1} \cdot \dots \cdot A_1 \\ \Psi_{i-1} = C_{i-1} \cdot [A_{i-1} \cdot \dots \cdot A_2 \cdot B_1, A_{i-1} \cdot \dots \cdot A_3 \cdot B_2, \dots, B_{i-1}] \\ \Omega_{i-1} = C_{i-1} \cdot [A_{i-1} \cdot \dots \cdot A_2, A_{i-1} \cdot \dots \cdot A_3, \dots, I] \end{cases}. \tag{10}$$

This can be solved applying least squares to the expression (9), considering $\bar{x}_0 = 0$.

$$\Psi_{i-1} \cdot U_i = -C_{i-1} \cdot \bar{x}_{i-1}. \tag{11}$$

## 4. Case Study

This case study has been developed to validate the proposed meta-model through the implementation of a simulation model. In this way, it is intended to exemplify the use of the meta-model developed as well as to highlight the potential applicability of some of the utilities supported by it. Likewise, this case study simulates an assembly line using different control alternatives to analyze and compare their effect from the perspectives of geometric quality and productivity. The approach of the case study and the results obtained are presented below.

### 4.1. Approach to the Case Study

The case presented is based on the process used in the study case of [10]: an assembly process of a 2D product composed of four parts which is assembled in a 3-stage manufacturing process and it is inspected at the end of the process, as illustrated in Figure 9.

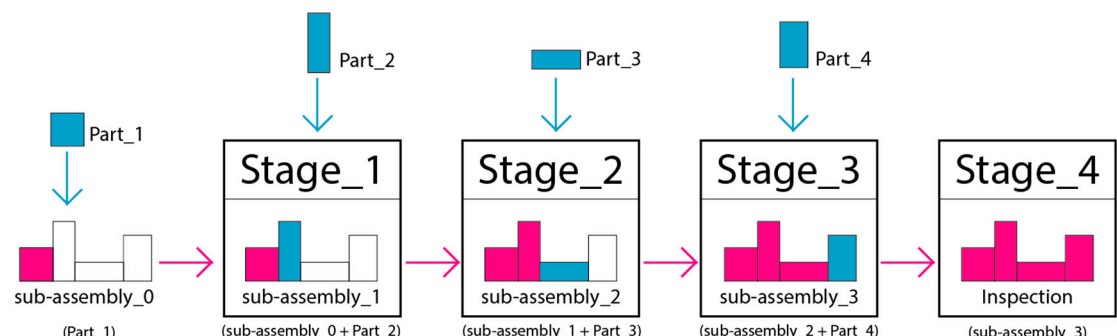

**Figure 9.** Case study process schema.

Three possible scenarios have been considered on this process depending on the type of control used:

- Scenario_0: No correction actions are applied. The VFS_Basic_Monitor&Control block monitors some of the variables of each stage and performs a statistical analysis of the final inspection of products carried out in Stage 4. Measurements of this final inspection are used to calculate the mean and the standard deviation of the geometric deviations obtained at each inspected point. In addition, the maximum geometric deviation and its corresponding inspection point is identified, as well as the number of products that do not meet specifications.
- Scenario_1: The in-process measuring capability of Stage_1 is used and, based on the measurement results, a VFS_FeedForward_Monitor&Control class calculates the optimal position of the locators in the subsequent stages for each assembly order to minimize the deviations obtained in the final inspection of each product. It must be also considered that the fixtures at Stages 2, 3, and 4 have actuators to adjust the position of each locator. For this reason, the setup times of these stages are modified to simulate the adjustment of the actuators.
- Scenario_2: the VFS_FeedForward&Backward_Monitor&Control block is used. In addition to the control actions considered in Scenario_1, corrections are calculated in the locators of Stage_1 using a statistical analysis of the measurements detected in the previous processed assemblies in Stage_1. For this reason, the setup time of Stage_1 is also modified to simulate the adjustment of the actuators.

Figure 10 graphically represents the two control alternatives considered.

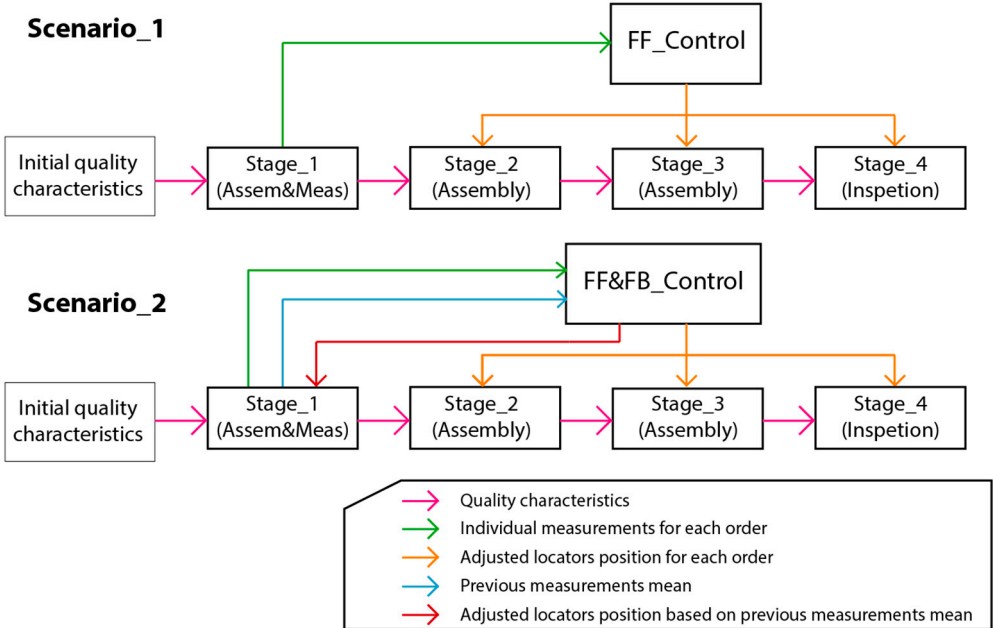

**Figure 10.** Scenario_1 and Scenario_2 process scheme.

However, both Scenario_1 and Scenario_2 require the use of actuators that adjust the position of the locators. In order to refine the productivity analysis of the proposed scenarios, it has been considered that the process can use automatic or manual actuators, with their respective consequences in setting times. Thus, a total of five alternative cases are obtained to study and compare:

- Case 1: Simulation of Scenario_0.
- Case 2: Simulation of Scenario_1 using manual actuators.
- Case 3: Simulation of Scenario_1 using automatic actuators.
- Case 4: Simulation of Scenario_2 using manual actuators.
- Case 5: Simulation of Scenario_2 using automatic actuators.

Some general parameters considered for the case study are presented below:

- The processing time means for each stage in Case_1 are: 200 s, 220 s, 180 s, and 80 s, respectively. These time means are maintained in the rest of the cases, except in Stage_1 which includes an in-process measurement; its new time mean is 240 s. A 10 s standard deviation is defined for every process time.
- The setup time for each stage with actuators is 80 s for manual actuators and 20 s for automatic actuators.
- Initial part deviations are modelled as a normal distribution with 0 mean and 0.05 mm standard deviation at coordinates X and Z. In addition, another pair of random deviations at coordinates X and Z are generated to deduce a random orientation deviation.
- Initial locator deviations are modelled as a normal distribution with 0 mean and 0.02 mm standard deviation at coordinates X and Z. However, a bias of the positioners has been modelled by introducing a progressive increase in the mean parameter of the locator deviation distribution, following a quadratic function. This mean parameter is restored to 0 every 3000 s to simulate their adjustment, repair, or replacement.
- The measurement noise for the Stage_1 (assembly station with in-process measurement) is defined by a normal distribution with 0 mean and 0.03 mm standard deviation.
- The measurement noise for the inspection stage is defined by a normal distribution with 0 mean and 0.01 mm standard deviation.

Figure 7 shows the implementation of the assembly line in OpenModelica, although some elements, such as the Control block, are replaced depending on the modelled scenario. To delve into other conditions considered in the case study, we suggest consulting [10], in which more detailed information of the assembly process and geometry data are presented. From these conditions, various simulations of the process have been run with each of the proposed scenarios in order to compare the various control alternatives and analyze their effect from the perspectives of geometric quality and productivity.

*4.2. Case Study Results*

The results obtained show that the control strategy proposed has notably improved the quality of the parts produced, observing a generalized decrease in the maximum deviations detected. Figure 11 shows the graphical representation of the mean values and the standard deviations of the maximum deviations detected at Stage_4. The dark blue line represents Case 1 data (Scenario_0). Red and purple lines represent Case 2 and 3, respectively, showing considerable improvements compared to Case 1. However, this improvement is even greater in Cases 4 and 5, represented with green and light blue lines, respectively.

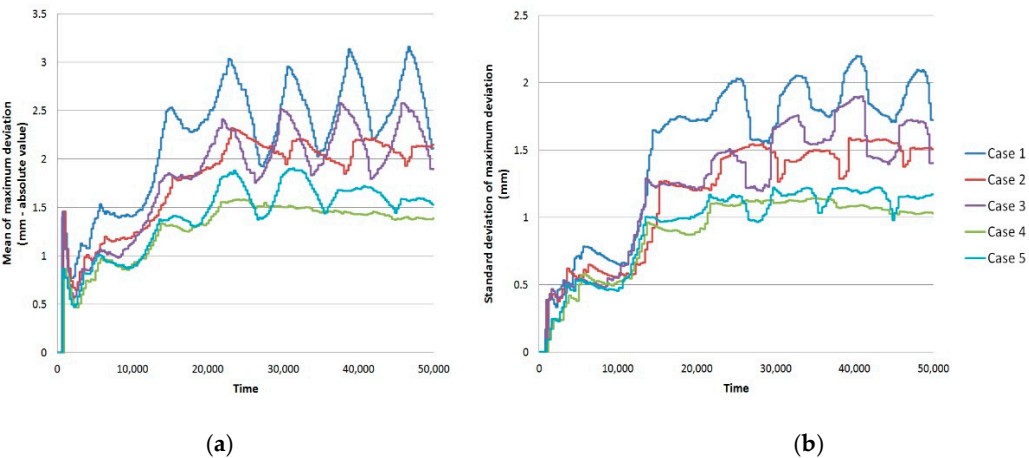

(**a**)                    (**b**)

**Figure 11.** Statistical analysis of maximum deviation detected at final inspection (Stage_4). (**a**) Mean; (**b**) standard deviation.

Considering the results related to productivity, every simulation has been run for the same simulation time (50,000 s) to compare the total number of finished products. Moreover, the throughput (finished products per hour) is also calculated. Likewise, with regard to the analysis of geometric quality, a limit of 3 mm (in absolute value) has been established for the maximum allowed deviation, obtaining the number of finished products that meet the specifications and being able to calculate the failure percentage with respect to total production (defect index). All these results are summarized in Table 1.

**Table 1.** Results table. Six results are summarized: finished products (F.P); finished products that meet specifications (FPMS); finished products per hour (throughput); defect index (D.I.); finished products that meet specifications per hour (FPMSH).

|  | **Case 1** | **Case 2** | **Case 3** | **Case 4** | **Case 5** |
|---|---|---|---|---|---|
| F.P. | 214 | 164 | 203 | 153 | 180 |
| Throughput | 15.41 | 11.81 | 14.61 | 11.01 | 12.96 |
| F.P.M.S. | 155 | 132 | 164 | 136 | 156 |
| D.I. | 27.57 | 19.51 | 19.21 | 11.11 | 13.33 |
| F.P.M.S.H. | 11.17 | 9.51 | 11.81 | 9.80 | 11.24 |

The main conclusions based on the analysis of the results of Table 1 are presented below:

- Case 1 presents the highest throughput. However, this case also has the highest D.I.
- The introduction of the feed-forward control (Cases 2 and 3) allows for improvement of the quality of the finished products and to reduce the D.I.
- In Cases 4 and 5, a backward control is also incorporated, observing a greater improvement in the quality results and obtaining the minimum D.I.
- The consideration of manual and automatic actuators in locator positioning considerably reduces the number of finished products. This effect is greater in the cases with manual actuators (Cases 2 and 4).
- In the cases where manual actuators are considered (Cases 2 and 4), the improvement in quality is not enough to offset the reduction of throughput.
- In Case 3, the results show that, despite reducing the total number of products produced, the improvement in quality causes an increase in the number of finished products that meet specifications, obtaining the best result.

As expected, the introduction of measurement and adjustment operations causes a reduction in the units produced. However, if we combine these data with the quality results, we can see that the total number of finished products that meet specifications is notably higher in Case 3 (automatic feed-forward control). However, it would be interesting to develop other types of analysis, such as a cost study, to cross-check with the results obtained and to be able to perform a more complete multi-domain analysis.

Beyond the analysis of the results of the case study, it is important to highlight that the implementation and simulation of the models for each scenario has been satisfactorily completed, obtaining results consistent with the conditions set out.

## 5. Discussion

This section is divided into two parts. First, we focus the discussion on the meta-model applied in the simulation prototype and the case study, and later on, we drive the attention to the approaches adopted and the proposed methodology from an industry 4.0 perspective and focus on geometric variability control.

Regarding the case study, the instantiated models and simulations to analyze different control logics for geometric variability reduction have shown different positive aspects:

- Validity of the meta-model. The correct construction, compilation, and simulation of the instances demonstrate the correct structuring and implementation of the meta-model. Moreover, it has been proven that the modular structure of the meta-model allows to build executable simulation models in a few minutes and in a very intuitive way.
- Multi-domain simulation approach. It has been proven that a multidomain global assessment is necessary to conduct an adequate analysis in MAS. For instance, according to the results in the best case scenario, the quality improvement increased the number of parts within specifications and the number of parts manufactured decreased with respect to other alternatives. Thus, an analysis to prioritize productivity would have provided a different alternative.
- Flexible simulation. The model instantiation supports the simulation of assembly lines with an interesting level of detail, allowing the introduction of various characteristics and behaviors of the process and obtaining realistic and reliable results. Moreover, the flexible and modular structure is aimed to facilitate the extension of the model to incorporate new block types, such as the consideration of new strategies (e.g., selective assembly) and logic control (e.g., based on establishment of optimal number and placement of inspection stations). For this reason, Modelica and the proposed libraries are a great potential resource for the analysis of assembly lines at the design stage.

On the other hand, the experience using the proposed methodology has proven that the use of SysML is a key element in the simulation system modelling since its block-based structure may

be rewritten quite easily into Modelica formalism. However, recent research works point out the need to advance in some aspects, such as the automatic validation of SysML models, the automatic transformation SysML–Modelica, etc. As these advances are reflected in the new versions of the standards published by the Object Management Group and Modelica groups, the industry will tend to adopt similar approaches to the one proposed in this work.

During this work, we have stated the lack of alignment of SysML diagrams to discrete manufacturing systems, as it has been reported [28]. Therefore, the development of normative SysML profiles that consider the modelling of manufacturing systems from different points of view (different types of analysis) and the development of standard libraries for these manufacturing entities is still an open issue. Furthermore, the normative developments on SysML should make some progress and include all aspects required for an integrated modelling process of a product-process-manufacturing system (e.g., MAS). This is pointed out in [29] due to the clear evidence of the strong interdependence that exists between a product and its production system, especially when dealing with geometric variability models during product, process, and manufacturing systems lifecycles. SysML and Modelica have shown their ability to represent simplified product geometries (lightweight models) and tolerances [21,30,31].

Another remarkable aspect of the SysML and Modelica languages that makes them suitable for our goals is that they promote the decomposition of complex systems into discrete components and subsystems that are later integrated into a system model, allowing composition of high-scale and low detail (lightweight) models, such as the one proposed in this paper and others of smaller scale and higher detail to form a virtual prototype and a digital twin. In the context of our research, this property is very important because it is possible to model the behavior of low-level components, such as those corresponding to the computer numerical control machine axis system, cutting tools, or manufacturing stages, and integrate them into global models. In this regard, it is of interest to mention that the SoV technique allows for modelling these low-level components to later model their interaction for analyzing the global behavior of the process, as proposed in [32,33], for example.

Finally, the importance of creating more flexible production system simulations within CPS and Digital Twin paradigms should be remarked. For this purpose, and according to [34], it is a key issue to model specific aspects and behaviors of the production system separately from the core simulation, including black-box modules to the main simulation model. The multidisciplinary co-simulation (multi-resolution simulation, hybrid simulation modelling, etc.) will assume an increasingly decisive role also for short-term decisions and at the phases of ramp-up and actual production [35], especially when dealing with scalable and highly modular production environments enabled by CPS.

## 6. Conclusions

In this work, a simulation meta-model has been developed to analyze MAS under the production quality paradigm, making use of system modelling tools, as advocated by the MBSE approach. Specifically, the proposed model integrates the unified study of the productive and geometric quality aspects of a MAS. DEVS formalism has been used to model the logistic flow, while the SoV technique has been used to model the variability propagation. Their coupled use has been feasible in allowing us to obtain valid executable simulation models.

Moreover, a methodology for the general development of simulation systems has been proposed. The methodology has been successfully applied, allowing to obtain a well-structured simulation model consistent with the MAS design to be simulated. Furthermore, it has allowed us to adopt a modular approach with the implementation of Modelica libraries. These libraries enable easy model instantiation and they show a promising capability for model expansion to support advanced manufacturing systems, such as a flexible and modular assembly line.

Using the initial SysML structural and behavioral modelling, the meta-model has been implemented successfully in OpenModelica for the development of executable simulation models, exploiting Modelica language capabilities for multi-domain co-simulations. Finally, the developed

Modelica libraries have been successfully instantiated to construct an executable model for a fictitious case study. The execution of this instantiated model has confirmed the meta-model's goodness for the study of system performance from an integrated perspective. In this case study, a control strategy based on locator adjustment has been simulated, comparing two control logics.

With all of this, it seems evident/clear that the development of tools and meta-models that support multi-domain simulations is a very interesting line of work in the field of manufacturing systems. Thus, our future research work will be focused on incorporating the behavior models of a smaller scale system (multi-axis systems, locator configuration systems with automatic adjustment, etc.) in order to achieve a more detailed and accurate quality simulation.

**Author Contributions:** Conceptualization, F.R.S. and P.R.C.; methodology, F.R.S.; software, S.B.N. and P.R.C.; validation, S.B.N., P.R.C., F.R.S. and J.V.A.-N.; investigation, S.B.N. and F.R.S.; writing—original draft preparation, S.B.N.; writing—review and editing, S.B.N., F.R.S. and J.V.A.-N.; supervision, F.R.S.; funding acquisition, S.B.N. All authors have read and agreed to the published version of the manuscript.

**Funding:** This research was funded by Generalitat Valenciana, grant number ACIF/2019/095.

**Conflicts of Interest:** The authors declare no conflict of interest.

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
