# Peer review of "Multidomain Simulation Model for Analysis of Geometric Variation and Productivity in Multi-Stage Assembly Systems"

_applsci, doi:10.3390/app10186606_

Round 1
Reviewer 1 Report
The topic of the manuscript is very actual and of great interest for industries such as automotive and aerospace. The results documented in the literature cover different aspects of multi-stage manufacturing systems (MMS), so the abstract needs to be improved by pointing out better the novelty of the proposed research. The recalled SoV analysis is recognized as significant to model MMS, and the advantages coming from modeling approach using SySML coupled with Modelica should be better highlighted.
In the introduction, when describing Modelica, a reference to that environment is appreciated. The same for SySML.
The methodology description is very detailed (making the full manuscript very long to read) and in some case it is difficult to follow if the logic behind SySML and the links/interactions among the several blocks of the MAS structure is not very familiar to the reader. The authors could try to simplify the description in section 2 and 3 addressing the reader to the main concepts of the methodology.
The equations (5) and (8) are not very clear and the included parameters need to be stated.
The final discussion could be shortened focusing the attention on the novelty of the obtained results and making a comparison with other ways to study MMS.
In the references, papers [1] and [13] are identical.
Author Response
Dear reviewer,
We would like to thank you for your review and the feedback you have provided. Likewise, we appreciate your comments and suggestions to improve the paper. As a result of all reviewers' comments, the following changes have been applied:
- the abstract has been extended to highlight the novelty of the proposal
- sections 2 and 3 have been shortened in order to improve readability. We are aware that the complexity of the proposal requires a certain specific knowledge of modeling but we want to highlight the contents described in these sections because they have been an important part of our work.
- discussion and conclusions sections have also been shortened to highlight the evaluation of the results obtained;
- All the text has been reviewed for improving the expression and readability, and some errors in bibliographic references have been fixed.
Below we detail the main changes made in the paper for responding to your particular comments:
The topic of the manuscript is very actual and of great interest for industries such as automotive and aerospace. The results documented in the literature cover different aspects of multi-stage manufacturing systems (MMS), so the abstract needs to be improved by pointing out better the novelty of the proposed research. The recalled SoV analysis is recognized as significant to model MMS, and the advantages coming from modeling approach using SySML coupled with Modelica should be better highlighted.
Response: we have extended the abstract in order to emphasize the novelty of the proposal. We have highlighted the advantages of the coupling of SysML and Modelica languages.
____________
In the introduction, when describing Modelica, a reference to that environment is appreciated. The same for SySML.
Response: We have incorporated two additional references to facilitate the non-expert reader’s introduction to SysML and Modelica.
____________
The methodology description is very detailed (making the full manuscript very long to read) and in some case it is difficult to follow if the logic behind SySML and the links/interactions among the several blocks of the MAS structure is not very familiar to the reader. The authors could try to simplify the description in section 2 and 3 addressing the reader to the main concepts of the methodology.
Response: Although we agree with the reviewer's perception, we have not been able to synthesize further the initial description of the methodology. We consider it a fundamental part of the work to value the adopted approach. However, we have removed various contents from sections 2 and 3 in order to facilitate reading and avoid an overly detailed discussion on some issues.
____________
The equations (5) and (8) are not very clear and the included parameters need to be stated.
Response: We have reformulated the text introducing equations (5) and (8) incorporating a reference to equations (1) and (2), from which the main parameters considered are extracted.
____________
The final discussion could be shortened focusing the attention on the novelty of the obtained results and making a comparison with other ways to study MMS.
Response: The discussion and conclusions sections have been also shortened in order to highlight the results of the case study. Furthermore, this change has entailed a reduction in the bibliographic references.
____________
In the references, papers [1] and [13] are identical.
Response: We thank you for noticing this error. We have eliminated it.
____________
Again, we appreciate all your comments and suggestions, and we hope that the suggested changes have responded to your indications in a satisfactory way.
Reviewer 2 Report
First of all, I would like to congratulate the authors for the clarity and quality of the work presented.
The title of the article suggests the topic.
The abstract reflects important aspects of the content of the article, emphasizing the need to develop a digital prototype to integrate productivity and part quality based on the Stream of Variation analysis in multistage 15 assembly systems.
The introduction provides a clear picture of the need to develop an emerging production in the future and of the issues related to improving and maintaining the quality of production. Various effective control strategies to prevent the generation and spread of defects throughout the manufacturing process and increase productivity have been demonstrated by applying the Stream of Variation (SoV) model or the Smart Manufacturing paradigm.
The work proposes is obviously - the development of a multi-domain and multi-scale modeling and simulation prototype that allows visualizing and evaluating the functionalities and operation modes of multi-stage assembly systems (MAS), both in terms of geometric quality and productivity.
Although the issue addressed in the article is relatively old and less addressed by researchers (modeling and simulating the effects of geometric variations for quality assurance, productivity and their interrelations) it remains current due to the need to develop new prototypes of multi- and multi-scale modeling and simulation prototype that allows visualizing and evaluating the functionalities and operation modes of multi-stage assembly systems (MAS), both in terms of geometric quality and
The organization of the article is a classic one, scientifically correct.
In section 2 the authors provide a clear picture of the methodologies and techniques used in this study, the SoV model is briefly described for MAS, defining the error propagation framework and the main sources of error in this type of MMS and the proposed methodology for designing the meta- model for MAS simulation.
Both - the modeling of error propagation through the SoV model (2.1) and the methodology adopted to define the simulation model (5 stages), (2.2.) are presented in detail, being accessible to most researchers and supported by multiple bibliographic notes.
Section 3 shows a logical thread of presentation and argumentation on: the need for flexible integration of quality control loops by implementing new control strategies focused on preventing the generation and spread of geometric defects; the need to validate a strategy with the construction of a simple simulation prototype supported by a meta-model and on the simulation of a MAS for the assembly of a single product; designing the prototype considering the Modelic capacity; the component of the "Simulation Assembly Process" block; the behavior model of the indivisible blocks governed by the DEVS atomic model, the implementation of the SysML meta-model in the Modelica language based on the textual code for defining classes and connectors with sufficient details to generate executable simulation models; VFS Basic Monitor & Control that performs the monitoring of the characteristic quality flow, which makes the article easy to understand and navigate.
Line 212 - I recommend text reformulation “reference in our research, in this work we focus on the simulation of a MAS”
The case study (Section 4) is explained in detail and the results obtained indicate that the proposed control strategy leads to a significant improvement in the quality of the parts produced, correlated with a general decrease in the maximum deviations detected.
The results (Section 5) highlight a modular structure of the meta-model that allows to build executable simulation models in minutes and in a very intuitive way, allowing quality improvement by increasing the number of parts within the specifications and decreasing the number of parts manufactured in in relation to the alternative ones.
The results were clearly presented in a logical sequence, with the authors performing an appropriate analysis.
The statements in section 6 (Conclusions) are supported by the results, being clear and reasonable. This research contributes practically and scientifically to the development of manufacturing systems in the future.
Author Response
Dear reviewer,
We would like to thank you for your review and the feedback you have provided. Likewise, we appreciate your comments and suggestions to improve the paper. As a result of all reviewers' comments, the following changes have been applied:
- the abstract has been extended to highlight the novelty of the proposal
- sections 2 and 3 have been shortened in order to improve readability. We are aware that the complexity of the proposal requires a certain specific knowledge of modeling but we want to highlight the contents described in these sections because they have been an important part of our work.
- discussion and conclusions sections have also been shortened to highlight the evaluation of the results obtained;
- All the text has been reviewed for improving the expression and readability, and some errors in bibliographic references have been fixed.
Below we detail the main changes made in the paper for responding to your particular comments:
Line 212 - I recommend text reformulation “reference in our research, in this work we focus on the simulation of a MAS”
Response: For improving the readability of paper, we have reformulated this sentence as follows: “For the sake of this simplicity, in this work we focus on the simulation of a MAS...”
_____________________
Again, we appreciate all your comments and suggestions, and we hope that the introduced changes attending other reviewer’s suggestions have not altered your good opinion about our paper.